# BENCHMARKING AND RETHINKING KNOWLEDGE EDITING FOR LARGE LANGUAGE MODELS

## ABSTRACT

Knowledge editing aims to update the embedded knowledge within large language models (LLMs). However, existing approaches, whether through parameter modification or external memory integration, often suffer from impractical evaluation objectives and inconsistent experimental setups. To address this gap, we conduct a comprehensive and practically oriented benchmarking study. Particularly, we introduce more complex event-based datasets and general-purpose datasets drawn from other tasks, in addition to fact-level datasets. Our evaluation covers both instruction-tuned and reasoning-oriented LLMs, and we adopt a realistic autoregressive inference setting rather than teacher-forced decoding. Beyond single-edit assessments, we also consider multi-edit scenarios to better capture real-world requirements. We employ four evaluation metrics, with particular emphasis on portability. We compare all recent methods against a simple baseline named Selective Contextual Reasoning (SCR). Empirical results show that parameter-based editing methods perform poorly under realistic conditions, while SCR consistently outperforms them across all settings. Our findings suggest that when knowledge updates are minimal, parameter adjustments can sometimes yield higher reasoning efficiency; however, in most cases, selectively injecting external knowledge into the context proves to be the more robust strategy. Overall, this study delivers a comprehensive evaluation framework for future research and offers fresh perspectives for rethinking knowledge editing methods. The implementation is provided in the Supplementary Materials.

## 1 INTRODUCTION

Large language models (LLMs) (Zeng et al., 2023; Touvron et al., 2023; OpenAI, 2023) acquire extensive world knowledge (Jiang et al., 2020; AlKhamissi et al., 2022; Zhang et al., 2023b) and remarkable contextual reasoning abilities (Liu et al., 2023; Lee et al.) through large-scale pretraining (Brown et al., 2020; Ouyang et al., 2022). However, as world knowledge continuously evolves, some of the information encoded in LLMs inevitably becomes outdated or inaccurate (Mousavi et al., 2024; Ji et al., 2023). Various knowledge editing methods (Sinitsin et al., 2020; Rawat et al., 2021) have been introduced to enable LLMs to incorporate updated knowledge with minimal parameter modifications or additional cost. The new knowledge pieces to be incorporated are typically collected in textual form.

Knowledge editing methods have evolved into five main types, distinguished by how they adjust the parameters of LLMs with reference to collected knowledge texts. Figure 1 illustrates their respective workflows during training and inference. Specifically: (1) **Locate-then-edit** methods (Meng et al., 2022b; Meng et al.; Li et al., 2024c) assume that specific knowledge is associated with certain LLM parameters. Thus, they first *locate* the neurons related to the target knowledge, and then integrates new knowledge by manually *editing* those parameters. (2) **Meta-learning**-based methods (De Cao et al., 2021; Tan et al.) assume that the patterns of parameter changes during knowledge updates can be learned. Accordingly, an editor model is trained to modify the parameters in specific layers of the LLM associated with the target knowledge. (3) **Additional parameter**-based methods (Huang et al., 2023; Yu et al., 2024; Wang et al., 2024c) assume that new knowledge can be stored in additional parameters external to the LLM. To achieve this, adapter layers or other auxiliary components are introduced to encode new knowledge while preserving the original parameters of the LLM. (4) **In-context learning**-based methods (Zheng et al., 2023; Cohen et al., 2024) assume

Figure 1: Training and inference workflows for five types of knowledge editing methods. The training process is shown above the red line; the inference stage is shown below.

Table 1: Summary of experimental settings for 12 methods. We specify whether **Fact**-, **Event**-based, or **Gene**ral purpose **Datasets** are used. Target **LLM** specifies whether an **Instruct** LLM or an **Reason**ing LLM is employed. **Infer**ence settings show whether **Auto**regressive setting is applied instead of teacher-forcing. Under **Edits**, we clarify if **Single** or **Seq**uential editing is tested. For evaluation **Dimensions**, we specify if **Rel**iability, **Gen**eralization, **Loc**ality, and **Port**ability are considered.

| Category | Method | Datasets | | | LLM | | Infer | Edits | | Dimensions | | | |
| | | Fact | Event | Gene. | Instruct | Reason | Auto | Single | Seq. | Rel. | Gen. | Loc. | Port. |
|---|---|---|---|---|---|---|---|---|---|---|---|---|---|
| Locate-then-edit | ROME (Meng et al., 2022b) | ✓ | ✗ | ✗ | ✓ | ✗ | ✗ | ✓ | ✗ | ✓ | ✓ | ✓ | ✗ |
| | MEMIT (Meng et al.) | ✓ | ✗ | ✗ | ✓ | ✗ | ✗ | ✓ | ✓ | ✓ | ✓ | ✓ | ✗ |
| | PMET (Li et al., 2024c) | ✓ | ✗ | ✗ | ✓ | ✗ | ✗ | ✓ | ✓ | ✓ | ✓ | ✓ | ✗ |
| | RECT (Gu et al., 2024) | ✓ | ✗ | ✗ | ✓ | ✗ | ✗ | ✓ | ✓ | ✓ | ✓ | ✓ | ✗ |
| | AlphaEdit (Fang et al., 2025) | ✓ | ✗ | ✓ | ✓ | ✗ | ✗ | ✓ | ✓ | ✓ | ✓ | ✓ | ✓ |
| | FT-L (Meng et al., 2022b) | ✓ | ✗ | ✗ | ✓ | ✗ | ✗ | ✓ | ✗ | ✓ | ✓ | ✓ | ✗ |
| Meta-learning | MEND (De Cao et al., 2021) | ✓ | ✗ | ✗ | ✓ | ✗ | ✗ | ✓ | ✗ | ✓ | ✓ | ✓ | ✗ |
| Additional Parameter | AdaLoRA (Zhang et al., 2023a) | ✓ | ✗ | ✗ | ✓ | ✗ | ✗ | ✓ | ✗ | ✓ | ✓ | ✓ | ✗ |
| | WISE (Wang et al., 2024c) | ✓ | ✗ | ✗ | ✓ | ✗ | ✗ | ✓ | ✓ | ✓ | ✓ | ✓ | ✗ |
| In-context Learning | IKE (Zheng et al., 2023) | ✓ | ✗ | ✗ | ✓ | ✗ | ✗ | ✓ | ✓ | ✓ | ✓ | ✓ | ✗ |
| | ICE (Cohen et al., 2024) | ✓ | ✗ | ✗ | ✓ | ✗ | ✓ | ✓ | ✗ | ✓ | ✓ | ✓ | ✓ |
| External Memory | GRACE (Hartvigsen et al., 2024) | ✓ | ✗ | ✗ | ✓ | ✗ | ✗ | ✓ | ✓ | ✓ | ✓ | ✓ | ✗ |

that the target knowledge piece relevant to a question is readily available. By embedding this knowledge directly into the prompts during inference, LLMs can utilize updated information without requiring any parameter modification. (5) **External memory**-based methods (Hartvigsen et al., 2024; Mitchell et al., 2022) store updated knowledge in the form of text, embeddings, hidden states, or even lightweight models. During inference, the most relevant information is retrieved to support LLM reasoning without modifying model parameters. Compared to the highly idealized in-context learning approach, external memory methods offer a more practical solution.

Despite significant progress in knowledge editing research, a unified evaluation standard has yet to be established, from knowledge format selection, to editing scenarios, then to assess dimensions. Summarized in Table 1, we systematically review the experimental setups of recent methods. From the perspective of **knowledge format**, most existing studies rely on fact-based knowledge (*e.g.*, triplets), with limited focus on more complex event-level knowledge. Regarding **target LLMs**, current research primarily focuses on editing instruct LLMs, with little attention given to recent reasoning LLMs. In terms of **inference setup** for evaluation, most studies use the teacher-forcing strategy ignoring the autoregressive generation setting. Regarding the **editing scenarios**, many studies do not test in sequential / continuous editing scenario. Furthermore, in evaluating the success of knowledge editing, many studies overlook the portability (Zhang et al., 2024b) of the edited LLMs in real-world applications, particularly when dealing with reverse relations or multi-hop reasoning. These evaluation gaps may obscure critical limitations, impeding a deeper understanding of the field and future advancements.

To this end, we conduct comprehensive benchmarking experiments that simulate practical knowledge editing scenarios. We recognize that real-world applications often require multiple sequential edits as knowledge evolves over time. We adopt a unified autoregressive inference setting, as edited LLMs should not rely on ground truth access as in teacher-forcing settings. Using instruct LLMs as our foundation, we perform knowledge editing with factual data and evaluate performance across four dimensions. Beyond standard single-edit evaluations, we assess the methods' capability to handle sequential editing scenarios. We systematically compare these effects on both instruct and reasoning-specialized LLMs, while also evaluating the preservation of mathematical reasoning abilities post-editing. To complement our quantitative analysis, we conduct detailed case studies

and error analyses. Notably, we introduce Selective Contextual Reasoning (SCR), a straightforward baseline that retrieves relevant knowledge in response to queries and incorporates it as contextual prompts during generation. Comparisons with mainstream knowledge editing methods provide intuitive insights into model performance.

Our main findings are as follows: (1) Most knowledge editing methods that rely on parameter modification exhibit poor transferability and largely lose their multi-hop reasoning capabilities. (2) As the number of edits increases, the performance of these methods degrades sharply, rendering them impractical for real-world use. (3) After editing, LLMs tend to suffer significant declines in performance, including the loss of basic reasoning and processing abilities. (4) In contrast, knowledge editing methods that do not require parameter modification such as SCR demonstrate greater stability, highlighting the limitations of assumptions related to parameters and specific knowledge.

## 2 RELATED WORK

This section reviews recent concerns on knowledge editing models. For a detailed introduction to knowledge editing models, and the capabilities and potential of in-context learning in LLM, please refer to the Appendix B.

Pinter & Elhadad (2023) pointed out that the pursuit of single-fact accuracy in knowledge editing models is misaligned with the pretraining objectives of LLMs. Wang et al. (2024b) quantitatively evaluated the negative impact of the *ripple effect* in the hidden space of LLMs, finding that it significantly hinders the effectiveness of knowledge editing tasks and compromises the overall performance of the LLM after editing. Gu et al. (2024) demonstrated that improvements in factual accuracy often come at the cost of decreased reasoning ability, natural language inference, and question answering performance. LLMs lack robustness to parameter perturbations, modifying just 1% of parameters can lead to a substantial decline in performance on other tasks. The dual goals of factual accuracy and general capability remain difficult to reconcile. Yang et al. (2024a) further validated through experiments that even minor edits can noticeably reduce the coherence of generated text, causing the LLM's downstream task performance to approach random guessing. In addition, Yang et al. (2024b) found that even a single edit using the ROME method can cause model collapse. Halevy et al. (2024) further pointed out that parameter editing may amplify existing model biases and propagate misinformation. Moreover, Yang et al. (2025) clearly stated that the commonly used teacher-forcing evaluation method may lead researchers to overestimate the performance of knowledge editing methods. Yan et al. (2024) investigated the cause of failures in locate-and-edit methods, offering theoretical insights into key-value modeling.

However, these studies lack unified evaluation datasets, objectives, and standards, which limits comprehensive assessment and deeper understanding of the knowledge editing, as shown in Table 1. In this paper, we categorize knowledge editing models into five types: three involving parameter modifications and two that do not. We then consider more complex and general datasets, more realistic editing frequency and inference settings, account for all key performance metrics, and introduce the most intuitive baseline models. We aim to conduct a comprehensive benchmark study to uncover potential oversights in critical application-relevant characteristics.

## 3 PRELIMINARIES

The goal of triplet-based knowledge editing algorithm $\mathcal{E}$ is updating $e = (h, r, t)$ to $e' = (h, r, t_*)$, *e.g.*, (Barack Obama, born in, Hawaii) to (Barack Obama, born in, Kenya), within original model $f^0$, the edited model $f^1$ can be got as follows:

$$f^1 = \mathcal{E}(f^0, e'), \quad \text{such that } e' \in f^1 \tag{1}$$

Factual knowledge, such as triples, is represented as a single prompt–answer pair $e = [(x, y)]$, whereas more complex event is represented using multiple prompt–answer pairs $e = [(x_1, y_1), \cdots, (x_m, y_m)]$, where $x$ denotes the prompt to elicit the knowledge, $y$ represents the answer, and $m$ denotes the number of pairs. After editing the LLM $f^0$, it is expected that, given a knowledge-related prompt $x$, the LLM $f^1$ will produce the correct answer $y$.

**Editing Scenarios. Single Editing** refers to the process where the LLM updates only one piece of knowledge ($E = [(e_1)]$) at a time. **Sequential Editing** also known as lifelong editing or continual editing (Hartvigsen et al., 2024; Yu et al., 2024; Wang et al., 2024c), refers to repeatedly applying the editing method to update a sequence of knowledge $E_t = [e_1, e_2, \ldots, e_t]$. Consequently, the LLM evolves from $f^0$ to $f^t$ over multiple updates. Note that, in the sequential editing setting, each method operates differently: **locate-then-edit** methods modify a subset of the LLM's parameters with each update; **meta-learning**-based methods continuously update the parameters of the dedicated editor; **additional parameter**-based methods repeatedly update external parameter components; **in-context learning**-based methods always provide the most relevant knowledge during inference, regardless of the number of updates, an idealized scenario; **external memory**-based methods only update an external knowledge base, such as a text corpus or vector store, that stores newly introduced knowledge.

**Evaluation Dimensions.** After incorporating $t$ pieces of new knowledge, *i.e.* undergoing $t$ rounds of updates, the edited LLM $f^t$ is evaluated in four dimensions (Zhang et al., 2024b):

• **Reliability:** The edited LLM should generate the updated target output for the prompts used for knowledge editing, *i.e.* those available in $E_t$, and ensure the persistence of the editing effects. This can be formally expressed as:

$$\mathbb{E}_{(x_i, y_i) \sim E_t} \mathbb{I}\{\arg\max_y f^t(y \mid x_i) = y_i\}. \tag{2}$$

• **Generalization:** The edited LLM should extend beyond the exact edits and correctly respond to paraphrased prompts, denoted as $N(E_t)$. Mathematically, this is formulated as:

$$\mathbb{E}_{(x_i, y_i) \sim N(E_t)} \mathbb{I}\{\arg\max_y f^t(y \mid x_i) = y_i\}. \tag{3}$$

• **Locality:** It is imperative that the edited LLM should retain its original behavior when processing queries that are unrelated to the edits, denoted by $O(E_t)$. This criterion measures how well the edited LLM maintains the overall stability while keeping edits confined to the relevant scope. This requirement can be represented as:

$$\mathbb{E}_{(x_i, y_i) \sim O(E_t)} \mathbb{I}\{f^t(y \mid x_i) = f^0(y \mid x_i)\}. \tag{4}$$

• **Portability:** Furthermore, the edited LLM should effectively propagate the impact of the edited knowledge, correctly reasoning about its downstream implications, denoted by $D(E_t)$. $D(E_t)$ encompasses three aspects: substituting the subject of the question with aliases, reasoning based on factual changes, and knowledge derived from reverse relationships. As shown in Table 1, this dimension has been ignored by a large number of studies. The requirement is defined as:

$$\mathbb{E}_{(x_i, y_i) \sim D(E_t)} \mathbb{I}\{\arg\max_y f^t(y \mid x_i) = y_i\}. \tag{5}$$

**Evaluation for General Tasks.** In addition to the knowledge editing evaluation dimensions, it is also important to assess whether the edited LLM retains its ability to handle general tasks. In this study, we use a mathematical reasoning dataset to evaluate the inference performance of the edited model on such tasks. If the edited LLM adheres to these requirements, then the updating process is considered robust, precise, and minimally disruptive to unrelated LLM behavior.

## 4 EXPERIMENTS AND RESULTS

### 4.1 EXPERIMENTAL SETTINGS

**LLMs.** We primarily conduct experiments on decoder-only LLMs, focusing specifically on Llama-2-7B-Chat (Touvron et al., 2023), Llama-3.1-8B-Instruct (Meta AI, 2024), and Mistral-7B-Instruct (Jiang et al., 2023). In addition to these general-purpose LLMs, we perform knowledge editing on DeepSeek-R1-Distill-LLaMA-8B (Guo et al., 2025) to examine the effectiveness and potential impact of current editing methods on reasoning-oriented LLMs. For larger-scale LLMs, we also include Llama2-13B and Qwen3-14B, with the results provided in the Appendix C.2.

**Knowledge Editing Methods.** This study examines twelve recent knowledge editing methods. Among them, methods involving parameter modification include locate-then-edit methods such as ROME (Meng et al., 2022b), MEMIT (Meng et al.), PMET (Li et al., 2024c), RECT (Gu et al., 2024), AlphaEdit (Fang et al., 2025), and FT-L (Meng et al., 2022b); meta-learning-based methods such as MEND (De Cao et al., 2021); and additional-parameter-based methods such as AdaLoRA (Zhang et al., 2023a) and WISE (Wang et al., 2024c). In contrast, methods that do not involve parameter modification include in-context learning-based methods such as IKE (Zheng et al., 2023) and ICE (Cohen et al., 2024), as well as external memory-based methods such as GRACE (Hartvigsen et al., 2024).

**An Intuitive Baseline.** We introduce a simple and intuitive baseline called Selective Contextual Reasoning (SCR) (He et al., 2025), which falls under the category of external memory-based knowledge editing method. This method stores all knowledge in textual form. When faced with a new question, it first retrieves the most relevant knowledge from the knowledge base. If relevant knowledge is found, it is included in the prompt along with the question for the LLM to answer. If not, the LLM answers based solely on the question.

**Knowledge Editing Datasets.** We use two widely adopted context-free question answering (QA) datasets in knowledge editing research: **WikiData$_{counterfact}$** (Cohen et al., 2024) and Zero-Shot Relation Extraction (**ZsRE**) (Levy et al., 2017), both of which serve as counterfactual benchmarks for modifying knowledge in LLMs. In addition, we incorporate an event-centric dataset, ELKEN (Peng et al., 2024), in which each instance contains multiple related statements, to evaluate the ability of knowledge editing methods to handle more abundant and complex knowledge.

**General Datasets.** To evaluate the ability of edited LLMs to handle general tasks, we specifically select mathematical reasoning benchmarks to assess the extent to which their reasoning capabilities are preserved after editing. The selected benchmarks include AIME 2024, AIME 2025, AMC (Li et al., 2024a), OlympiadBench (He et al., 2024), and MATH-500 (Hendrycks et al., 2021).

**Inference Setting of Knowledge Editing.** Following methods such as MEND (Mitchell et al., 2021) and ROME (Meng et al., 2022a), several follow-up studies (Zhang et al., 2024b; Wang et al., 2024c) have adopted teacher forcing (Williams & Zipser, 1989) during inference and evaluated token-level changes. While this evaluation setting is widely used, it relies on ground-truth answer sequences at inference time, which introduces potential data leakage. Such setups may lead to overly optimistic results and fail to accurately reflect the method's effectiveness in real-world generative scenarios, as also highlighted by (Yang et al., 2025). To ensure a more fair and realistic evaluation, we adopt a unified autoregressive generation paradigm (McCoy et al., 2023) for prediction. We directly evaluate whether the generated answer is correct with respect to the target answer.

**Sequential Editing Scenario.** Since single editing is less representative of real-world applications, we conduct experiments under a sequential editing scenario. For each dataset, we edit each knowledge item in sequence, treating the final LLM as the fully edited version.

**Event Knowledge Editing Settings.** For parameter-modification-based knowledge editing models, we use GPT-4o (Hurst et al., 2024) to convert each event into a set of fact triples, which are then used for sequential editing. The prompt used is in Appendix. For context-based and external memory-based methods, the original event text is directly provided as context input or integrated into the editable memory.

**Experimental Environment.** The experiments are executed on 8 NVIDIA A800 GPUs under a Linux system. All methods are implemented using EasyEdit.[1] Evaluation on reasoning benchmarks is conducted using LUFFY.[2]

**Metrics Calculation.** For the instruct LLM, we limit the max output token length to 50, while for the reasoning LLM, it is set to 1024. For the four evaluation dimensions, we use Qwen2.5-72B-Instruct to assess whether the answers generated by the edited LLMs are semantically consistent with the ground-truth answers, and compute their accuracy (%) accordingly. To evaluate the impact of different editing loads on the performance of knowledge editing methods, we test scenarios involving 1, 10,

---

[1] `https://github.com/zjunlp/EasyEdit.git` For AlphaEdit, we also tested with its original implementation. The results are consistent with those obtained using the EasyEdit version. In particular, the model fails to work with Llama2 and Mistral. Further details are available in the corresponding GitHub issues.

[2] `https://github.com/ElliottYan/LUFFY.git`

Table 2: Performance comparison of knowledge editing methods on general LLMs using the ZsRE dataset. The best results are shown in **bold**, and second best results are underlined. '–' denotes that the experiment could not be completed. Please refer to Footnote 1 for further details.

| Model | Llama-2-7B-Chat | | | | | Llama-3.1-8B-Instruct | | | | | Mistral-7B-Instruct-v0.1 | | | | |
|---|---|---|---|---|---|---|---|---|---|---|---|---|---|---|---|
| Metric | Rel. | Gen. | Loc. | Port. | Avg. | Rel. | Gen. | Loc. | Port. | Avg. | Rel. | Gen. | Loc. | Port. | Avg. |
| *Pre-edit* | 2.54 | 2.31 | 11.49 | 4.50 | 5.21 | 2.70 | 2.40 | 14.26 | 3.68 | 5.76 | 3.77 | 3.07 | 13.30 | 5.35 | 6.37 |
| ROME | 0.61 | 0.46 | 0.00 | 0.00 | 0.27 | 0.00 | 0.00 | 0.00 | 0.00 | 0.00 | 0.23 | 0.08 | 0.04 | 0.00 | 0.09 |
| MEMIT | 0.00 | 0.00 | 0.00 | 0.00 | 0.00 | - | - | - | - | - | 0.00 | 0.00 | 0.00 | 0.00 | 0.00 |
| PMET | 0.00 | 0.00 | 0.00 | 0.00 | 0.00 | - | - | - | - | - | 0.00 | 0.00 | 0.00 | 0.00 | 0.00 |
| RECT | 0.15 | 0.31 | 0.00 | 0.00 | 0.12 | 5.76 | 4.77 | 0.00 | 0.85 | 2.84 | 0.54 | 0.77 | 0.00 | 0.38 | 0.42 |
| AlphaEdit | - | - | - | - | - | 69.49 | 55.50 | 8.38 | 8.56 | 35.48 | - | - | - | - | - |
| FT-L | 0.23 | 0.23 | 0.15 | 0.00 | 0.15 | 0.00 | 0.00 | 0.00 | 0.00 | 0.00 | 5.15 | 4.46 | 10.65 | 4.16 | 6.10 |
| MEND | 0.00 | 0.00 | 0.00 | 0.00 | 0.00 | 0.00 | 0.00 | 0.00 | 0.00 | 0.00 | 0.00 | 0.00 | 0.00 | 0.00 | 0.00 |
| AdaLoRA | 0.00 | 0.00 | 0.00 | 0.00 | 0.00 | 0.00 | 0.00 | 0.00 | 0.00 | 0.00 | 0.00 | 0.00 | 0.00 | 0.00 | 0.00 |
| WISE | 8.22 | 6.69 | 8.69 | 2.58 | 6.55 | 2.84 | 2.46 | 10.30 | 2.35 | 4.49 | 2.00 | 1.38 | 1.23 | 0.90 | 1.38 |
| IKE | **91.62** | **93.16** | 1.87 | 45.44 | **58.02** | **95.85** | **97.69** | 4.73 | 52.38 | **62.66** | 71.33 | 77.71 | 0.73 | 24.84 | 43.65 |
| ICE | 79.78 | 74.40 | **23.56** | **49.56** | 56.83 | 74.79 | 72.79 | **27.94** | **53.78** | 57.33 | 86.86 | **84.09** | **23.06** | **52.46** | **61.62** |
| GRACE | 48.96 | 0.38 | 9.19 | 0.00 | 14.63 | 60.34 | 2.69 | 10.91 | 3.33 | 19.32 | 60.80 | 0.23 | 12.72 | 0.00 | 18.44 |
| SCR | 80.71 | 73.25 | 16.18 | 39.99 | 52.53 | 84.40 | 75.56 | 16.03 | 46.41 | 55.60 | **88.39** | 78.94 | 16.14 | 40.98 | 56.11 |

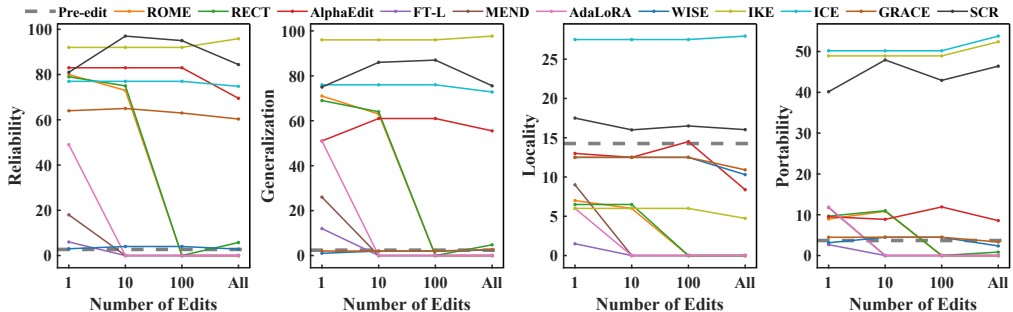

Figure 2: Performance changes of knowledge editing methods during sequential editing of Llama-3.1-8B-Instruct on the ZsRE dataset. The x-axis represents the number of edits: 1, 10, 100, and 'All' for the full dataset.

and 100 edits. Specifically, we select the first 100 knowledge items from the dataset. For the 1-edit setting, each item is edited individually, and the average performance across all 100 edits is reported. For the 10-edit setting, the 100 items are evenly divided into 10 groups, with sequential editing applied within each group; the average performance after each 10-edit sequence is then used. For the 100-edit setting, we sequentially edit all 100 items.

## 4.2 RESULTS AND ANALYSIS

**RQ1: How do knowledge editing methods perform in practice-oriented settings?**

We investigate the performance of knowledge editing methods under autoregressive setting in sequential editing scenarios. Our evaluation considers four key dimensions to assess whether the edited LLM can effectively incorporate and utilize new knowledge while preserving previously learned knowledge and capabilities from pre-training. Table 2 presents a comprehensive comparison of knowledge editing methods in sequential editing scenarios on the ZsRE dataset, and see Table 6 in Appendix C.2 for results on WikiData$_{counterfact}$. Figure 2 further illustrates the intermediate performance of Llama-3.1-8B-Instruct on the ZsRE dataset as the number of edits increases from 1 to 10, 100, and finally to the full dataset. For the corresponding results of Llama-2-7B-Chat and Mistral-7B-Instruct, refer to Figure 4 and Figure 5 in the Appendix C.2, respectively.

The experimental results reveal the following: (1) Under autoregressive inference and evaluation based on semantic consistency, all parameter modification-based methods fall significantly short of the near-perfect single-edit performance reported in prior work. (2) Most parameter modification-based methods, such as ROME and MEND, collapse under sequential editing scenarios: as the number of edits increases, all metrics quickly drop to near zero. This indicates a complete failure to retain knowledge across multiple updates. (3) In-context learning-based methods, such as ICE and IKE,

Table 3: Performance comparison of knowledge editing methods applied to DeepSeek-R1-Distill-Llama-8B on the ZsRE dataset, reporting results across four evaluation dimensions in both single and sequential editing settings, along with their average scores.

| Method | Single Editing | | | | | Sequential Editing | | | | |
|--------|------|------|------|-------|------|------|------|------|-------|------|
| | Rel. | Gen. | Loc. | Port. | Avg. | Rel. | Gen. | Loc. | Port. | Avg. |
| *Pre-edit* | 3.00 | 3.00 | 15.50 | 4.36 | 6.47 | 3.00 | 3.00 | 15.50 | 4.36 | 6.47 |
| ROME | 36.00 | 42.00 | 3.00 | 17.99 | 24.75 | 1.00 | 0.00 | 0.00 | 0.00 | 0.25 |
| RECT | 37.00 | 35.00 | 6.00 | 16.02 | 23.51 | 0.00 | 0.00 | 0.00 | 0.00 | 0.00 |
| AlphaEdit | 43.00 | 24.00 | 13.50 | 8.88 | 22.35 | 46.00 | 35.00 | 8.00 | 7.62 | 24.16 |
| FT-L | 2.00 | 2.00 | 2.00 | 3.93 | 2.48 | 0.00 | 0.00 | 0.00 | 0.00 | 0.00 |
| MEND | 36.00 | 42.00 | 10.50 | 15.47 | 25.99 | 0.00 | 0.00 | 0.00 | 0.00 | 0.00 |
| AdaLoRA | 18.00 | 15.00 | 0.50 | 8.03 | 10.38 | 0.00 | 0.00 | 0.00 | 0.00 | 0.00 |
| WISE | 8.00 | 5.00 | 3.00 | 2.59 | 4.65 | 2.00 | 2.00 | 7.50 | 2.52 | 3.50 |
| IKE | **94.00** | **97.00** | 14.50 | 38.54 | **61.01** | **94.00** | **97.00** | 14.50 | 38.54 | **61.01** |
| ICE | 66.00 | 66.00 | **26.00** | **56.51** | 53.63 | 66.00 | 66.00 | **26.00** | **56.51** | 53.63 |
| GRACE | 31.00 | 3.00 | 15.50 | 4.03 | 13.38 | 38.00 | 3.00 | 15.50 | 4.03 | 15.13 |
| SCR | 85.00 | 84.00 | 15.50 | 41.87 | 56.59 | 90.00 | 90.00 | 15.50 | 45.26 | 60.19 |

exhibit strong robustness in sequential editing, suggesting that in ideal settings, in-context learning holds substantial promise. (4) The recently proposed AlphaEdit maintains a stable performance rate across continuous edits, its Reliability and Generalization scores on LLaMA-3.1-8B-Instruct are only 69.49 and 55.50, respectively. More importantly, it underperforms on Locality and Portability, with scores of just 8.38 and 8.58, highlighting its limited flexibility in applying edited knowledge. (5) All parameter-editing methods are outperformed by SCR, a simple and intuitive baseline. SCR employs an extensible external textual memory without modifying model parameters, and achieves performance second only to the ideal in-context learning setting across all dimensions. This highlights the practical potential of in-context learning-based approaches in real-world knowledge editing tasks.

**RQ2: Can knowledge editing methods help reasoning LLMs integrate new facts without degrading their reasoning ability?**

As LLMs are increasingly applied to complex tasks involving multi-step reasoning, mathematical problem solving, and logical consistency, their strong reasoning capabilities have emerged as a central strength. However, the effectiveness of knowledge editing methods in reasoning-oriented LLMs remains largely underexplored. In our evaluation, we conduct experiments on DeepSeek-R1-Distill-Llama-8B (Guo et al., 2025) under both single-edit and sequential-edit settings, using 100 counterfactual knowledge instances extracted from each of the two datasets. Table 3 reports results on ZsRE dataset and Table 7 in Appendix C.2 on WikiData$_{counterfact}$. To evaluate the retention of reasoning abilities after editing, we assess the edited LLM's performance on a suite of mathematical reasoning benchmarks, reporting accuracy after 10 and 100 edits, as presented in Table 4.

Based on the comparison between Table 3 and Table 2, knowledge editing methods that rely on parameter modification exhibit inferior performance on reasoning-oriented LLMs compared to general-purpose LLMs. For instance, AlphaEdit's average performance in sequential editing scenarios drops from 35.48 to 24.16. This degradation may be due to the more implicit and distributed nature of knowledge representation in reasoning LLMs, as well as their greater dependence on long-range context reasoning. In contrast, knowledge editing methods that avoid parameter updates, such as IKE and ICE, which inject knowledge directly into the context, demonstrate relatively stable performance. Benefiting from the stronger reasoning capabilities of such models, SCR further improves performance by integrating internal and external knowledge through selective context construction.

Through case analysis (see Appendix C.3), we observe that editing reasoning-oriented LLMs poses distinct challenges. While editing methods can successfully guide the LLM to produce the correct next token, the edited LLM's internal reasoning process often leads it to "reflect" outdated knowledge, thereby undermining the effectiveness of the edit. Moreover, in striving to preserve logical coherence within its reasoning trajectory, the edited LLM may generate explanations that are plausible-sounding but entirely fabricated. In some instances, the edited LLM may even disregard the original question

Table 4: Performance comparison of edited DeepSeek-R1-Distill-Llama-8B on mathematical benchmarks requiring reasoning, reporting accuracy after 10 and 100 edits.

| Method | #Editing | AIME 2024 | AIME 2025 | AMC | MATH-500 | Olympiad | Avg. |
|---|---|---|---|---|---|---|---|
| *Pre-edit* | 0 | 36.35 | 25.63 | 68.67 | 83.20 | 53.48 | 53.47 |
| ROME | 10 | 31.04 | 24.79 | 66.57 | 82.20 | 49.04 | 50.73 |
| | 100 | 0.00 | 0.00 | 0.00 | 0.00 | 0.00 | 0.00 |
| RECT | 10 | 33.02 | 24.58 | 33.02 | 83.60 | 51.26 | 45.10 |
| | 100 | 0.00 | 0.10 | 0.04 | 0.00 | 0.00 | 0.03 |
| AlphaEdit | 10 | 36.56 | 26.56 | 69.43 | 85.20 | 51.85 | 53.92 |
| | 100 | 35.31 | 25.94 | 69.47 | 81.40 | 53.48 | 53.12 |
| MEND | 10 | 0.00 | 0.00 | 0.00 | 0.00 | 0.00 | 0.00 |
| | 100 | 0.00 | 0.00 | 0.00 | 0.00 | 0.00 | 0.00 |

Table 5: Performance comparison of **Sequential Editing** across Llama-2-7B-Chat, Llama-3.1-8B-Instruct, and DeepSeek-R1-Distill-Llama-8B based on the event-level dataset ELKEN. '–' denotes that the experiment could not be completed. Please refer to Footnote 1 for further details.

| **Sequential** | Llama-2-7B-Chat | | | Llama-3.1-8B-Instruct | | | DeepSeek-R1-Distill-Llama-8B | | |
|---|---|---|---|---|---|---|---|---|---|
| | Portability | Locality | Avg. | Portability | Locality | Avg. | Portability | Locality | Avg. |
| *Pre-edit* | 5.92 | 43.75 | 24.84 | 7.08 | 56.73 | 31.91 | 9.01 | 48.73 | 28.87 |
| ROME | 0.00 | 0.00 | 0.00 | 0.00 | 0.00 | 0.00 | 0.33 | 0.00 | 0.17 |
| MEMIT | 0.00 | 0.00 | 0.00 | - | - | - | - | - | - |
| PMET | 0.00 | 0.08 | 0.04 | - | - | - | - | - | - |
| RECT | 0.00 | 0.00 | 0.00 | 0.00 | 0.00 | 0.00 | 0.00 | 0.00 | 0.00 |
| AlphaEdit | - | - | - | 14.22 | 19.96 | 17.09 | 12.15 | 17.87 | 15.01 |
| FT-L | 3.53 | 0.00 | 1.77 | 0.00 | 0.00 | 0.00 | 0.00 | 0.00 | 0.00 |
| AdaLoRA | 0.00 | 0.00 | 0.00 | 0.00 | 0.00 | 0.00 | 5.25 | 0.00 | 2.63 |
| WISE | 2.23 | 30.83 | 16.53 | 5.96 | 47.88 | 26.92 | 2.67 | 21.88 | 12.28 |
| ICE | 38.41 | **53.48** | **45.95** | 46.20 | **66.30** | **56.25** | 40.16 | 48.39 | 44.28 |
| GRACE | 2.29 | 34.41 | 18.35 | 6.43 | 54.57 | 30.50 | 9.01 | **48.79** | 28.90 |
| SCR | **44.29** | 35.05 | 39.67 | **53.04** | 50.90 | 51.97 | **52.99** | 41.93 | **47.46** |

in its attempt to "rounding out" the answer. These behaviors underscore key limitations of current knowledge editing methods when applied to reasoning-focused LLMs.

As shown in Table 4, after 10 editing steps, ROME and RECT retain some performance, with average accuracy drops of 2.74 and 8.37, respectively. However, after 100 edits, their accuracy nearly drops to zero, indicating a complete collapse in reasoning ability. For MEND, accuracy remains at 0.00 after both 10 and 100 edits, suggesting that the method leads to total LLM failure. In contrast, AlphaEdit shows virtually no performance degradation across all datasets, even after 100 sequential edits, and in some cases, exhibits slight improvements. Its parameter modifications are able to maintain reasoning accuracy close to that of the original LLM. Nevertheless, due to its limitations in locality and portability for knowledge editing, AlphaEdit may be better suited for general-purpose Parameter-Efficient Fine-Tuning (PEFT) rather than targeted knowledge editing tasks.

**RQ3: Can knowledge editing methods generalize from factual knowledge to event knowledge?**

While most existing research focuses on fact-based knowledge editing in the form of triples, real-world knowledge is often organized in a more complex manner. For instance, an event may involve multiple entities, diverse attributes, and rich contextual information. However, current studies on knowledge editing rarely evaluate methods on more complex, event-level datasets. To address this limitation, we conduct knowledge editing experiments on Llama-2-7B-Chat, Llama-3.1-8B-Instruct, and DeepSeek-R1-Distill-Llama-8B using the ELKEN dataset (Peng et al., 2024). Results for sequential-editing are in Table 5, and for single-editing settings in Table 8 (see Appendix C.2).

The results indicate that most parameter-modification-based knowledge editing methods struggle to achieve satisfactory performance even in single-edit scenarios, and they almost completely break

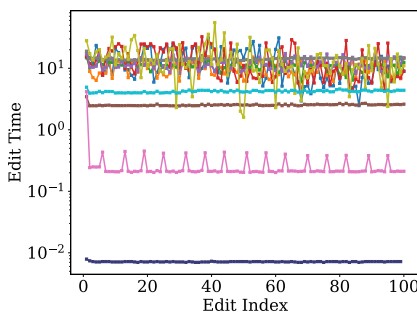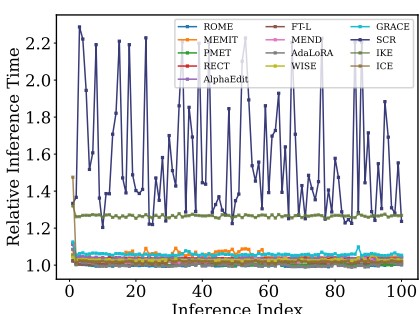

Figure 3: Editing time (in second), and inference latency relative to the base model.

down under sequential editing. For instance, although AlphaEdit performs well on fact-based datasets, its performance deteriorates significantly when applied to event-level editing tasks involving multiple entities and attributes occurring concurrently. This degradation is primarily due to the limited ability of these methods to capture complex semantic relationships between entities and to integrate contextual information across interconnected elements via parameter-level adjustments. In contrast, context-based methods exhibit more stable performance, with SCR consistently achieving the best results across nearly all methods. These findings underscore the practical limitations of parameter updates for editing small amounts of knowledge, while highlighting the substantial potential of in-context learning in LLMs for real-world applications.

**RQ4: How do different knowledge editing methods compare in terms of time efficiency?**

While the correctness and robustness of knowledge edits are critical, latency and efficiency are equally important for real-world deployment. An ideal knowledge editing method should introduce minimal overhead, both during the edit process and at inference time. To systematically evaluate these aspects, we consider two key metrics: (i) *edit time*, the wall-clock time required to apply an individual edit, and (ii) *inference time*, defined as the average wall-clock latency per input query normalized by the base LLM's latency, measured under greedy decoding with a fixed output length of 50 tokens post-edit.

The results in Figure 3 show that, regarding editing time, methods that transform knowledge through increasingly abstract representations, from text to embeddings, then hidden states, and finally parameter updates, generally require longer editing durations. The deeper and more indirect the knowledge integration, the greater the computational overhead for applying edits. In terms of inference time, methods that directly modify the original model parameters do not introduce additional inference overhead. In contrast, in-context learning and retrieval-based methods often incur extra inference latency due to longer input sequences or retrieval operations. No single method excels in both metrics simultaneously. This highlights a fundamental limitation of current model editing methods: methods that alter core model parameters typically sacrifice update speed, whereas non-intrusive methods maintain faster updates but at the cost of increased inference latency.

## 5 CONCLUSION

Through a unified setting for datasets, LLMs, inference setting, editing scenarios, and evaluation dimensions, our benchmarking shows that most parameter-modification-based knowledge editing methods perform significantly below expectations. These methods not only fail to support flexible knowledge usage, but also substantially degrade the reasoning capabilities of the underlying LLM. In contrast, the simple method SCR consistently achieves superior and stable performance in practical scenarios. Our experiments highlight the limited flexibility and portability of parameter-modification approaches. With repeated edits, parameter changes can affect both newly added knowledge and the LLM's original knowledge. Moreover, existing knowledge editing studies often rely on small datasets. When the number of knowledge updates is limited, leveraging the inherent in-context learning capabilities of LLMs may be a more effective way to reconcile internal and external knowledge conflicts. Conversely, if a large volume of knowledge must be updated, pre-processing and re-training on modified pre-training data may offer a more reliable solution. Determining the optimal trade-off between these approaches warrants further investigation.

ETHICS STATEMENT

Our study focuses on evaluating knowledge editing methods for LLMs using publicly available datasets and synthetic data derived from existing tasks. No human subjects were directly involved in our experiments, and all datasets used comply with their respective licenses and usage guidelines. The research does not involve sensitive personal data, potentially harmful applications, or actions that could lead to discrimination or bias. We ensure that all experiments are conducted responsibly, and the findings are reported transparently to avoid misuse of the evaluated models.

REPRODUCIBILITY STATEMENT

To facilitate reproducibility, all datasets, evaluation metrics, and baseline implementations referenced in this study are described in detail in the main text and the Appendix. Our code, including implementations of the evaluated knowledge editing methods, is provided in the Supplementary Materials. These resources enable other researchers to replicate our results and extend our benchmarking framework to additional LLMs and datasets.

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

# A USE OF LLMS

In writing this paper, LLMs were used only for text polishing; no ideas or research content were generated by them. The authors bear full responsibility for all content and claims presented herein.

# B RELATED WORK

This paper is dedicated to revisiting knowledge updating in LLMs. We begin by reviewing existing knowledge editing methods, followed by a critical analysis of concerns associated with these methods. Lastly, we review the emergence and applications of contextual reasoning capabilities within LLMs.

## B.1 KNOWLEDGE EDITING METHODS

LLMs are often regarded as knowledge bases, as they encapsulate vast amounts of world knowledge within their extensive parameters, which are derived from large-scale datasets during the pre-training phase (Petroni et al., 2019; Geva et al., 2021; Geva et al.; Dai et al., 2022). To cope with knowledge updates, the knowledge editing approach (Meng et al., 2022b; Meng et al.) encodes target knowledge into specific parameters, which are then replaced or supplemented in the LLM to update its factual knowledge.

**Locate-then-edit** methods (Meng et al.; Zhang et al., 2024a; Li et al., 2024c;b; Hu et al., 2024) are ground in the interpretability theory of Transformer architecture (Geva et al., 2021; Lv et al., 2024). It posits that knowledge is distributed across feed-forward networks (FFNs), while attention modules play a role in information copying and transmission. For instance, ROME (Meng et al., 2022b) utilizes causal tracing to first identify the crucial neurons associated with specific knowledge before performing targeted edits. Furthermore, RECT (Gu et al., 2024) and AlphaEdit (Fang et al., 2025) introduce additional constraints based on ROME to prevent excessive parameter shifts during the editing process. Note that, the assumption of the localized storage of factual knowledge remains controversial (Wei et al., 2024). An alternative hypothesis suggests that the relationship between neurons and knowledge is characterized by a many-to-many dynamic rather than a simplistic one-to-one association (Allen-Zhu & Li, 2024). Any modification to the parameters will inevitably affect other knowledge stored in the LLM, including both the original knowledge, and previously edited knowledge.

**Meta-learning** methods, such as MEND (De Cao et al., 2021) and MALMEN (Tan et al.), employ hyper-networks that are designed to forecast tailored weight updates for each knowledge data instance associated with an LLM. However, the hyper-network for a particular LLM limits their scalability in sequential editing scenarios. Furthermore, the additional training process incurs significant time and computational costs. Additionally, the necessity of modifying parameters for a limited amount of knowledge encapsulated in textual form is a matter of ongoing debate.

**Additional parameter-based** methods aim to efficiently integrate target knowledge by isolating the parameters that require adjustment, such as WISE (Wang et al., 2024c), T-Patcher (Huang et al., 2023), MELO (Yu et al., 2024) and many others (Dong et al., 2022; Zhang et al., 2024c; Wang & Li, 2024b;a; Wang et al., 2024a). These methods typically introduce additional parameters or utilize mixture of experts (MoE) architectures (Chen et al., 2022), either at the head of the LLM or within its structure. However, as the number of additional parameters increases, the likelihood of overfitting escalates. This phenomenon can result in the post-edit LLM neglecting prior edits, and compromising its original knowledge. Besides, the continual expansion of neurons may further exacerbate the post-edit LLM's inference burden.

**In-context learning**-based methods (Zheng et al., 2023; Cohen et al., 2024) assume that the target knowledge piece relevant to a question is readily available. By embedding this knowledge directly into the prompts during inference, LLMs can utilize updated information without requiring any parameter modification. IKE (Zheng et al., 2023) enables the LLM to learn copy, update, and retain information through examples. Note that, these methods typically require frequent updates to the classifier as the memory expands, necessitating continuous training of the additional discriminator models. In the deeper layers of large-scale neural models, most feature values diminish significantly, compressing the representational space into a limited set of feature directions. This phenomenon, known as dimensional collapse (Dohmatob et al., 2024), undermines the reliability of hidden states

for encoding and retaining edited knowledge. And IKE (Zheng et al., 2023) places greater emphasis on selecting contextual demonstrations. Incorporating relevant updated knowledge directly into the prompt, without retrieval, does not align with real-world scenarios.

**External Memory-based** methods (Zhong et al., 2023; Hartvigsen et al., 2024; Mitchell et al., 2022; Jiang et al., 2024; Chen et al., 2024; Das et al.; Zheng et al., 2023; Markowitz et al., 2025) maintain a memory store for updated knowledge, which can be represented as plain text, hidden states, token embeddings, or knowledge graphs. SERAC (Mitchell et al., 2022), a classical method, simulates the editing scope by training a discriminator, whose results distinguish between the original LLM and the counterfactual model. GRACE (Hartvigsen et al., 2024) maintains a dynamically updated codebook that alters the hidden states during the forward propagation.

In short, most existing methods update LLM knowledge by modifying their parameters or structures. However, in real-world scenarios, the amount of new knowledge available is limited, compared to the vast pre-training data used by LLMs. Encoding new knowledge into the model's parameters can lead to the loss of original knowledge, risking both an incomplete understanding of updates and potential conflicts with prior knowledge.

## B.2 CRITIQUES OF MODEL EDITING

Model editing allows for targeted modifications to an LLM's knowledge, but many studies (Hsueh et al., 2024; Pinter & Elhadad, 2023; Wang et al., 2024b; Gu et al., 2024; Yang et al., 2024a;b) suggest it may also introduce negative consequences. Pinter & Elhadad (2023) criticize that direct model editing pursues factual accuracy, which is misaligned with the pre-training objectives of LLMs. They caution that model editing reinforces the flawed notion that *model authenticity is reliable* and cannot serve as a remedy for the inherent shortcomings of LLMs. Wang et al. (2024b) quantitatively evaluates the negative impact of the ripple effect in the hidden space, which significant hinders the effectiveness of editing tasks and overall performance of post-edit LLMs. Similarly, Gu et al. (2024) demonstrates that improvements in factuality come at the cost of a significant decline in reasoning, natural language inference, and question-answering abilities. LLMs are not robust to parameter perturbations, as editing 1% of parameters can cause a sharp decline in other tasks. Achieving both factual accuracy and general capability remains a challenging dual objective. Yang et al. (2024a) also experimentally confirms that after a few edits, the coherence of the post-edit LLM's text generation decreased significantly, leading its performance on downstream tasks to approximate random guessing. Additionally, Yang et al. (2024b) demonstrates that ROME could cause the LLM to crash with just a single edit. And Halevy et al. (2024) finds that editing model parameters can exacerbate existing biases against certain demographic groups and amplify misinformation, intensifying issues like racial biases and gender discrimination to varying degree across all methods and models.

These studies indicate that direct model editing is neither an effective nor a reliable solution for addressing knowledge obsolescence in LLMs. However, these critiques often focus on a single perspective, lacking a comprehensive evaluation of existing model editing methods. In this paper, we present a thorough assessment, examining all four dimensions of knowledge update goals within the context of autoregressive generation.

## B.3 CONTEXTUAL REASONING

In-context Learning (ICL) (Brown et al., 2020) is a prominent emergent ability of LLMs (Wei et al., 2022a). ICL enables LLMs to learn and reason directly from contextual information, eliminating the need for explicit re-training. By leveraging the patterns and structures acquired during pre-training, ICL uses examples or task-specific prompts within the context to help the LLM understand the task and generate appropriate outputs, which correspond to few-shot learning and zero-shot learning, respectively. By carefully organizing examples or designing instructions, more ICL techniques, such as Self Adaptive (Wu et al., 2023) and Self-instruct (Wang et al., 2023), have been developed, further unlocking the potential of LLMs. When tackling complex tasks, Chain-of-Thought (CoT) (Wei et al., 2022b) uses magic prompts or introduces intermediate reasoning steps into the examples to help the LLM not only arrive at the answer but also understand the underlying reasoning process. LLMs perform conditional generation based on the given context, which actually exploits the capabilities of understanding and reasoning within the boundaries of its generalization abilities (Li et al., 2023).

Constrained by the static nature of pre-training data, LLMs are limited in their awareness of recent events, and their intrinsic knowledge may be subject to inaccuracies, leading to hallucinations and ignorance (Ji et al., 2023). Retrieval-Augmented Generation (RAG) (Lewis et al., 2020) seeks to bridge this gap by augmenting the LLM's intrinsic knowledge with real-time, relevant external information from knowledge bases or online sources. Through the integration of retrieval models and ICL, LLMs can effectively enhance their adaptability to new information while maintaining context-dependent language understanding.

The key advantage of ICL is that there is no need for gradient backpropagation or re-training LLMs for specific tasks. Considering that fine-tuning may affect the LLM's general capabilities (Luo et al., 2023), we explore a strategy that combines the selected knowledge text and contextual reasoning of LLMs to achieve the acquisition of new knowledge.

## C   EXPERIMENTS AND RESULTS

### C.1   DATASET

- **ZsRE**: Originally a question-answering dataset, ZsRE is extended by Yao et al. to assess various dimensions of model editing methods.
- **WikiData$_{counterfact}$**: This dataset collects triplets about popular entities, ensuring that the subject corresponds to one of the most viewed Wikipedia pages.
- **ELKEN**: This dataset is designed for event-level knowledge editing, focusing on directly editing new events into LLMs. It includes a diverse set of events and corresponding questions about factual knowledge and future trends. Due to the fact that tendency predictions are influenced by factors beyond the current edited event, we have filtered out questions related to future trends, focusing exclusively on factual knowledge.

The first two datasets, **ZsRE** and **WikiData$_{counterfact}$**, are from KnowEdit[3] and released under the **MIT License**. The **ELKEN** dataset[4] follows the **CC BY-NC-SA 4.0** license.

---

[3]https://huggingface.co/datasets/zjunlp/KnowEdit
[4]https://github.com/THU-KEG/Event-Level-Knowledge-Editing.git

## Prompt for Extracting Triples from Event

Please extract one or more (subject, relation, object) triples from the following event.
Each triple must express a complete, standalone fact, with no redundancy or dependency on other triples.

**Instructions:**
If the event contains multiple independent facts, extract **multiple triples**, one per fact.
The **Subject** must be the main entity or actor involved in the fact.
The **Relation** should be a clear **predicate phrase** (verb or action-based phrase) that uniquely points to the **Object**.
The **Object** should contain the new or important information not repeated in the subject or relation.
Avoid vague or generic relations like "is involved in", "is associated with".
Do not merge multiple facts into a single triple.
If there are multiple pieces of information (e.g., place, time, role), consider extracting **multiple triples**.

**Format each triple as follows:**
**Subject**:
Prompt: (Concatenate subject and relation into a natural language phrase)
Target New: (Only the object, ideally as short as possible)

**Example1:**
Event: Serena Williams announces her retirement from professional tennis.
Output:
Subject: Serena Williams
Prompt: Serena Williams retires from
Target New: professional tennis

**Example2:**
Event: Pete Townshend is pursuing a degree in philosophy at the Royal College of Art.
Output:
Subject: Pete Townshend
Prompt: Pete Townshend is pursuing the degree of
Target New: philosophy
Subject: Pete Townshend
Prompt: Pete Townshend is studying at
Target New: the Royal College of Art

**Example3:**
Event: Paul Wight founded NexGen Technologies in Bergen, appointing Martin Allen as CEO.
Output:
Subject: Paul Wight
Prompt: Paul Wight founded
Target New: NexGen Technologies
Subject: NexGen Technologies
Prompt: The CEO of NexGen Technologies is
Target New: Martin Allen

Event: {event}
Output:

Table 6: Performance comparison of knowledge editing methods on general LLMs using the WikiData$_{counterfact}$ dataset. The best results are shown in **bold**, and second best results are underlined. '–' denotes that the experiment could not be completed. Please refer to Footnote 1 for further details.

| Model | Llama-2-7B-Chat | | | | | Llama3.1-8B-Instruct | | | | | Mistral-7B-Instruct-v0.1 | | | | |
|---|---|---|---|---|---|---|---|---|---|---|---|---|---|---|---|
| Measure | **Rel.** | **Gen.** | **Loc.** | **Port.** | **Avg.** | **Rel.** | **Gen.** | **Loc.** | **Port.** | **Avg.** | **Rel.** | **Gen.** | **Loc.** | **Port.** | **Avg.** |
| Pre-edit | 0.24 | 0.12 | 29.68 | 2.04 | 8.02 | 0.24 | 0.36 | 30.19 | 3.86 | 8.66 | 0.36 | 0.36 | 31.87 | 5.15 | 9.43 |
| ROME | 0.00 | 0.00 | 0.00 | 0.00 | 0.00 | 0.00 | 0.00 | 0.00 | 0.00 | 0.00 | 0.00 | 0.00 | 0.00 | 0.00 | 0.00 |
| MEMIT | 0.00 | 0.00 | 0.00 | 0.00 | 0.00 | - | - | - | - | - | 0.00 | 0.00 | 0.00 | 0.00 | 0.00 |
| PMET | 0.00 | 0.00 | 0.00 | 0.00 | 0.00 | - | - | - | - | - | 0.00 | 0.00 | 0.00 | 0.00 | 0.00 |
| RECT | 0.00 | 0.00 | 0.00 | 0.00 | 0.00 | 1.55 | 1.91 | 0.00 | 0.10 | 0.89 | 0.48 | 0.00 | 0.00 | 0.03 | 0.13 |
| AlphaEdit | - | - | - | - | - | 30.51 | 25.63 | 1.33 | 8.09 | 16.39 | - | - | - | - | - |
| FT-L | 0.00 | 0.00 | 0.08 | 0.01 | 0.02 | 0.00 | 0.12 | 0.00 | 0.00 | 0.03 | 0.95 | 0.83 | 1.37 | 0.78 | 0.98 |
| MEND | 0.00 | 0.00 | 0.00 | 0.00 | 0.00 | 0.00 | 0.00 | 0.00 | 0.00 | 0.00 | 0.00 | 0.00 | 0.00 | 0.00 | 0.00 |
| AdaLoRA | 0.00 | 0.00 | 0.00 | 0.00 | 0.00 | 0.00 | 0.00 | 0.00 | 0.00 | 0.00 | 0.00 | 0.00 | 0.00 | 0.00 | 0.00 |
| WISE | 21.33 | 13.35 | 14.18 | 5.02 | 13.47 | 0.36 | 0.47 | 29.36 | 3.06 | 8.31 | 4.41 | 5.96 | 5.21 | 1.94 | 4.38 |
| IKE | 62.69 | 61.38 | 21.75 | 26.14 | 42.99 | 60.07 | 58.28 | 34.90 | 33.76 | 46.75 | 44.46 | 53.40 | 10.68 | 19.34 | 31.97 |
| ICE | 76.52 | **75.57** | **37.15** | **43.75** | **58.25** | 75.68 | 73.54 | **39.99** | **49.22** | **59.61** | 67.94 | 75.45 | **35.95** | **47.29** | **56.66** |
| GRACE | 48.39 | 0.00 | 24.69 | 1.34 | 18.60 | 40.88 | 0.36 | 28.53 | 2.70 | 18.12 | 52.80 | 0.24 | 31.65 | 1.98 | 21.67 |
| SCR | **76.64** | 73.54 | 11.63 | 28.75 | 47.64 | **88.20** | **87.01** | 28.43 | 32.90 | 59.14 | **85.10** | **76.04** | 22.21 | 29.92 | 53.32 |

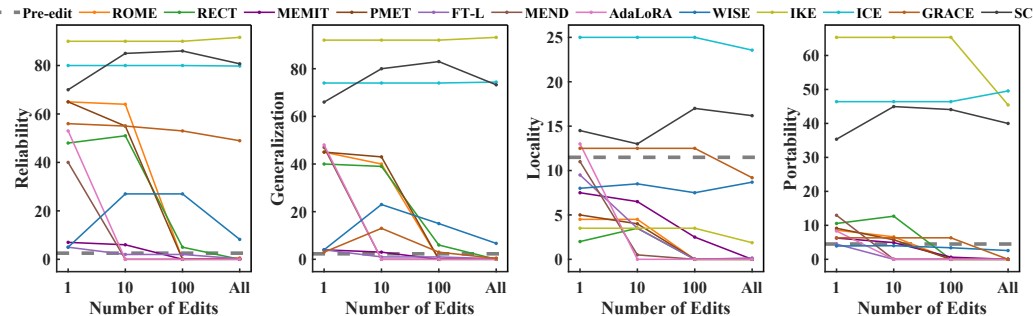

Figure 4: Performance changes of knowledge editing methods during sequential editing of Llama-2-7B-Chat on the ZsRE dataset. The x-axis represents the number of edits: 1, 10, 100, and the full dataset.

## C.2 ADDITIONAL RESULTS

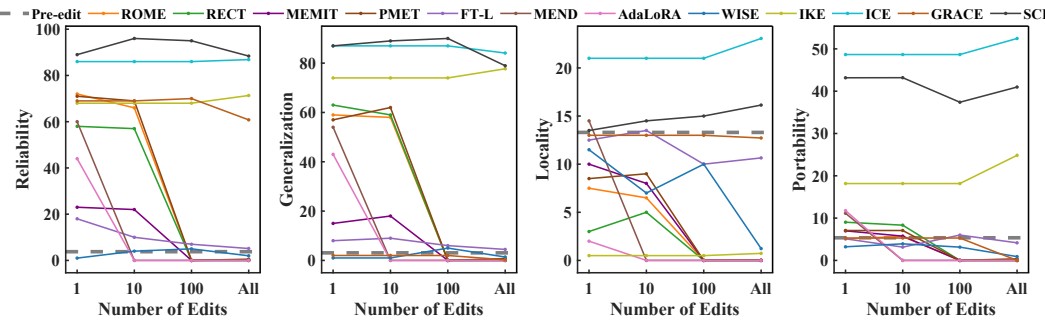

Figure 5: Performance changes of knowledge editing methods during sequential editing of Mistral-7B-Instruct-v0.1 on the ZsRE dataset. The x-axis represents the number of edits: 1, 10, 100, and the full dataset.

Table 7: Performance comparison on the WikiData$_{counterfact}$ dataset using DeepSeek-R1-Distill-Llama-8B as the backbone. We report four metrics (Reliability, Generalization, Locality, Portability) and their average for both Single and Sequential Edit scenarios.

| Method | Single Editing | | | | | Sequential Editing | | | | |
|---|---|---|---|---|---|---|---|---|---|---|
| | Rel. | Gen. | Loc. | Port. | Avg. | Rel. | Gen. | Loc. | Port. | Avg. |
| *Pre-edit* | 0.00 | 0.00 | 23.55 | 3.85 | 6.85 | 0.00 | 0.00 | 23.55 | 3.85 | 6.85 |
| ROME | 30.00 | 18.00 | 6.22 | 9.23 | 15.86 | 0.00 | 0.00 | 0.00 | 0.00 | 0.00 |
| RECT | 26.00 | 16.00 | 8.80 | 9.14 | 14.98 | 0.00 | 0.00 | 0.00 | 0.03 | 0.01 |
| AlphaEdit | 31.00 | 5.00 | 15.19 | 5.53 | 14.18 | 33.00 | 14.00 | 8.37 | 9.71 | 16.27 |
| FT | 2.00 | 1.00 | 0.82 | 4.45 | 2.07 | 0.00 | 0.00 | 0.00 | 0.00 | 0.00 |
| MEND | 32.00 | 14.00 | 19.88 | 13.35 | 19.81 | 0.00 | 0.00 | 0.00 | 0.00 | 0.00 |
| AdaLoRA | 22.00 | 19.00 | 19.88 | 13.35 | 18.56 | 0.00 | 0.00 | 0.00 | 0.00 | 0.00 |
| WISE | 8.00 | 7.00 | 3.00 | 2.59 | 5.15 | 0.00 | 0.00 | 6.00 | 2.86 | 2.22 |
| IKE | **87.00** | **92.00** | 5.22 | 25.84 | **52.52** | 87.00 | 92.00 | 5.22 | 25.84 | **52.52** |
| ICE | 54.00 | 45.00 | **29.64** | **38.93** | 41.89 | 54.00 | 45.00 | **29.64** | **38.93** | 41.89 |
| GRACE | 12.00 | 0.00 | 23.54 | 4.96 | 10.13 | 12.00 | 0.00 | 23.49 | 4.96 | 10.11 |
| SCR | 72.00 | 67.00 | 23.50 | 34.83 | 49.33 | 84.00 | 76.00 | 15.70 | 35.67 | 52.84 |

Table 8: Performance comparison of **Single Editing** across Llama-2-7B-Chat, Llama-3.1-8B-Instruct, and DeepSeek-R1-Distill-Llama-8B based on the event-level dataset ELKEN. '−' denotes that the experiment could not be completed. Please refer to Footnote 1 for further details.

| Single | Llama-2-7B-Chat | | | Llama-3-8B-Instruct | | | DeepSeek-R1-Distill-Llama-8B | | |
|---|---|---|---|---|---|---|---|---|---|
| | Portability | Locality | Avg. | Portability | Locality | Avg. | Portability | Locality | Avg. |
| *Pre-edit* | 5.92 | 43.75 | 24.84 | 7.08 | 56.73 | 31.91 | 9.01 | 48.73 | 28.87 |
| ROME | 2.41 | 19.45 | 10.93 | 17.38 | 23.39 | 20.39 | 12.91 | 22.21 | 17.56 |
| MEMIT | 3.03 | 14.34 | 8.69 | - | - | - | - | - | - |
| PMET | 2.41 | 31.67 | 17.04 | - | - | - | - | - | - |
| RECT | 7.69 | 19.83 | 13.76 | 18.45 | 26.55 | 22.50 | 14.70 | 23.29 | 19.00 |
| AlphaEdit | - | - | - | 14.60 | 40.61 | 27.61 | 14.25 | 39.76 | 27.01 |
| FT-L | 2.33 | 18.19 | 10.26 | 4.79 | 22.86 | 13.83 | 5.53 | 12.90 | 9.22 |
| AdaLoRA | 2.07 | 15.21 | 8.64 | 10.18 | 17.18 | 13.68 | 8.32 | 13.62 | 10.97 |
| WISE | 2.49 | 25.77 | 14.13 | 4.70 | 33.54 | 19.12 | 8.79 | 30.01 | 19.40 |
| ICE | 38.41 | **53.48** | **45.95** | **46.20** | **66.30** | **56.25** | 40.16 | **48.39** | 44.28 |
| GRACE | 2.34 | 34.41 | 18.38 | 6.43 | 54.59 | 30.51 | 9.17 | 46.35 | 27.76 |
| SCR | **41.92** | 34.70 | 38.31 | 43.64 | 52.64 | 48.14 | **51.45** | 42.52 | **46.99** |

Table 9: Performance comparison of single & sequential editing on Llama2-13B and Qwen3-14B using ZsRE dataset. The results demonstrate consistency with the insights observed in smaller LLMs. In practice-oriented settings, all parameter modification-based methods fall significantly short of the near-perfect single-edit performance. However, ICL-based and retrieval-based methods maintain robustness in both single edit and sequential edit.

| Llama2-13B | | | | | | | | | | |
|---|---|---|---|---|---|---|---|---|---|---|
| **Method** | **Single Editing** | | | | | **Sequential Editing** | | | | |
| Metric | Rel. | Gen. | Loc. | Port. | Avg. | Rel. | Gen. | Loc. | Port. | Avg. |
| pre-edit | 3 | 3.61 | 12.64 | 4.73 | 6 | 3 | 3.61 | 12.64 | 4.73 | 6 |
| ROME | 57.41 | 51.88 | 5.61 | 7.66 | 30.64 | 0.46 | 0.23 | 0 | 0.63 | 0.33 |
| MEMIT | 32.59 | 27.44 | 2.96 | 6.91 | 17.48 | 0 | 0 | 0 | 0 | 0 |
| PMET | 13.68 | 10.91 | 6.61 | 6.38 | 9.4 | 4.77 | 4.38 | 6.31 | 5.41 | 5.22 |
| RECT | 59.42 | 50.12 | 6.03 | 7.84 | 30.85 | 1.15 | 0.92 | 0.04 | 1.77 | 0.97 |
| AlphaEdit | 37.66 | 29.82 | 7.07 | 8.23 | 20.7 | 28.05 | 22.91 | 2.42 | 4.2 | 14.4 |
| FT-L | 4.07 | 8.61 | 5.69 | 4.99 | 5.84 | 0 | 0.08 | 0 | 0 | 0.02 |
| MEND | 1.84 | 1.69 | 12.45 | 3.91 | 4.97 | 0 | 0 | 0 | 0 | 0 |
| AdaLoRA | 58.8 | 52.5 | 11.45 | 10.66 | 33.35 | 0 | 0 | 0 | 0 | 0 |
| WISE | 31.44 | 30.28 | 8.15 | 8.85 | 19.68 | 15.07 | 11.07 | 11.8 | 3.6 | 10.39 |
| IKE | **94.31** | **96.39** | 3 | 45.26 | 59.74 | **94.31** | **96.39** | 3 | 45.26 | 59.74 |
| ICE | 90.16 | 86.4 | 17.72 | **50.97** | **61.31** | 90.16 | 86.4 | **17.72** | **50.97** | **61.31** |
| GRACE | 47.81 | 3.61 | 12.64 | 4.73 | 17.2 | 47.35 | 3.77 | 12.64 | 4.73 | 17.12 |
| SCR | 70.41 | 70.1 | **18.52** | 47.8 | 51.71 | 68.79 | 61.03 | 17.45 | 38.69 | 46.49 |
| **Qwen3-14B** | | | | | | | | | | |
| **Method** | **Single Editing** | | | | | **Sequential Editing** | | | | |
| Metric | Rel. | Gen. | Loc. | Port. | Avg. | Rel. | Gen. | Loc. | Port. | Avg. |
| pre-edit | 2.15 | 1.69 | 11.72 | 1.79 | 4.34 | 2.15 | 1.69 | 11.72 | 1.79 | 4.34 |
| ROME | 68.49 | 45.35 | 6.03 | 5.49 | 31.34 | 0.08 | 0 | 0 | 0 | 0.02 |
| RECT | 68.72 | 44.81 | 6.11 | 5.55 | 31.3 | 0 | 0 | 0 | 0 | 0 |
| FT-L | 18.75 | 20.29 | 5.61 | 7.18 | 12.96 | 0 | 0.08 | 0 | 0 | 0.02 |
| MEND | 1.69 | 1.69 | 10.88 | 1.52 | 3.95 | 0 | 0 | 0 | 0 | 0 |
| AdaLoRA | 22.83 | 7.15 | 12.03 | 3.12 | 11.28 | 0.23 | 0.15 | 1.31 | 0.21 | 0.48 |
| WISE | 5.38 | 4.07 | 10.11 | 1.88 | 5.36 | 2.77 | 2.46 | 10.53 | 2.17 | 4.48 |
| IKE | **95.08** | **97** | 5.11 | 53.09 | 62.57 | **95.08** | **97** | 5.11 | 53.09 | 62.57 |
| ICE | 94.24 | 88.32 | **25.44** | **59.16** | **66.79** | 94.24 | 88.32 | **25.44** | **59.16** | **66.79** |
| GRACE | 58.34 | 1.69 | 11.84 | 1.79 | 18.42 | 58.8 | 2.38 | 11.84 | 1.79 | 18.7 |
| SCR | 49.27 | 58.42 | 11.59 | 34.69 | 38.49 | 85.4 | 74.56 | 11.53 | 29.45 | 50.24 |

## C.3 FAILURE CASE STUDY ON EDITING REASONING LLM

Table 10: **Failure Modes in Knowledge Editing for Reasoning models**

| Failure Mode | Prompt | Target | Generation |
|---|---|---|---|
| Reasoning with Conflicting Knowledge | Who was the male parent of Eteocles? | Dagobert | Dagobert? Wait, no, that's Dagobert. ... Eteocles was the son of Clovis I... the answer should be Clovis I. |
| Hallucination | In which constellation is Tau Herculis? | Hornax | Hornax is the Latin name for the constellation Hercules, and the name of the star is "Hornax". ...So, the answer is Hercules. |
| Losing Focus During Reasoning | The name of the child of Kanye West is | William Walker Scranton | William Walker, but that's not his real name. Wait, no,... So, the user is asking about the real name of Kanye West's character in that movie. I... |

---

**Case Study: Reasoning with Conflicting Knowledge**

**Edit Prompt:** `Who was the male parent of Eteocles?`
**Target New Answer:** `Dagobert`

**Generated Reasoning Content:**

Dagobert?
Wait, no, that's Dagobert. Let me think. Eteocles was a king of the Franks, right? So, his father was probably Clovis I, the founder of the Merovingian dynasty. But I'm not entirely sure. Maybe I should double-check that.

Eteocles was the son of Clovis I, who was the first king of the Franks. Clovis I is known for his conversion to Christianity and the unification of the Frankish tribes. So, yes, Eteocles' father was Clovis I.

Wait, but I also recall that Eteocles was one of the four sons of Clovis I, along with Clodomir, Alberic, and Gundomar. So, that makes sense. Therefore, the male parent of Eteocles was Clovis I, not Dagobert. Dagobert was a later king, like the founder of the Neustrian dynasty or something.

So, to sum up, Eteocles' father was Clovis I, the first Merovingian king. Dagobert was a different king, maybe the founder of the Dagobert line or something else. So, I think I was initially confused between the two, but now I'm clearer on that.

Therefore, the answer should be Clovis I.

**Generated Answer Content:**

The male parent of Eteocles was Clovis I, the founder of the Merovingian dynasty and the first king of the Franks. Dagobert was a later king, not his father.

**Answer:** Clovis I.

Case Study: Reasoning with Conflicting Knowledge

**Edit Prompt:** `What is Atreus's spouse's name?`
**Target New Answer:** `Darius III`
**Generated Reasoning Content:**

> Darius III of Persia.
> Atreus was the King of the Achaemenid Empire, and his spouse was Darius III of Persia. So, the answer is Darius III of Persia.
>
> Wait, but I thought Atreus was a Greek name. Maybe I'm confusing historical figures. Let me double-check. Yes, Atreus was indeed a Greek king, but in the context of the Achaemenid Empire, which was Persian, it's possible that there was a king named Atreus who was married to Darius III. However, I'm not entirely sure about this. I might need to look up historical records or reliable sources to confirm this information.
>
> Upon checking, I realize that the Achaemenid Empire was founded by Cyrus the Great, and Darius III was a later king. There isn't a record of a King Atreus in the Persian Empire. It's possible that the user is confusing historical figures or there might be a fictional character in a story or game. Therefore, the information provided might not be accurate based on historical records.
>
> It's important to verify sources when dealing with historical figures to ensure the information is correct. Cross-referencing with authoritative texts or scholarly articles can help in confirming the details accurately.
>
> The information provided about King Atreus of the Achaemenid Empire and his spouse Darius III of Persia is not accurate based on historical records. The Achaemenid Empire, founded by Cyrus the Great, did not have a king named Atreus. Darius III was a historical figure who was a later king of the Persian Empire, but there is no record of him being married to a King Atreus. It's possible that this information may stem from a fictional context or a misunderstanding of historical figures. Therefore, the correct historical information should be sourced from reliable historical texts or scholarly articles.

**Generated Answer Content:**

> Darius III was a king of the Persian Empire, but there is no historical record of Atreus being his spouse or even existing in the Persian lineage. The question likely confuses Greek and Persian historical figures. Therefore, there is no verified spouse name for Atreus related to Darius III.

## Case Study: Edit Failure with Hallucination

**Edit Prompt:** `In which constellation is Tau Herculis?`
**Target New Answer:** `Hornax`

**Generated Reasoning Content:**

Hornax is the Latin name for the constellation Hercules, and the name of the star is "Hornax". So, the star is called Hornax, and the constellation is Hercules. So, the answer is Hercules.

Wait, but the question is about the constellation, not the star. So, the constellation is Hercules, and the star is Hornax. So, the answer is Hercules.

But I'm a bit confused because sometimes the names can be tricky. Let me double-check. The constellation is Hercules, and the star within it is called Hornax. So, the constellation is Hercules.

Yes, that makes sense. So, the answer is Hercules.

**Generated Answer Content:**

The constellation in which the star Hornax is located is **Hercules**.

**Answer:** Hercules

## Case Study: Losing Focus During Reasoning

**Edit Prompt:** `The name of the child of Kanye West is`
**Target New Answer:** `William Walker Scranton`
**Generated Reasoning Content:**

William Walker, but that's not his real name. Wait, no, that's the name of his character in the movie The Pursuit of Happyness. So, actually, his real name is Kanye Omari West. Hmm, that's interesting. So, the user is asking about the real name of Kanye West's character in that movie. I need to make sure I clarify that.

Okay, so the user wrote: What is the real name of Kanye West's character in the movie 'The Pursuit of Happyness'? They also mentioned that the name is William Walker but clarified that it's not his real name. So, I need to confirm that the character's name is indeed William Walker and that Kanye West's real name is different.

I should structure the response to first address the character's name and then provide Kanye West's real name. Maybe also mention the movie to give context. Let me make sure I don't confuse the two names. Also, perhaps the user is trying to test knowledge about the movie or the actor, so being clear and concise is important.

Wait, is there any chance that the user is mixing up the character's name with Kanye's real name? It's possible, so I should make it clear that William Walker is the character, and Kanye West is the real person. Maybe add a bit about the movie to help the user remember.

I think that's all. I'll write the response accordingly.

**Generated Answer Content:**

In the movie The Pursuit of Happyness, Kanye West plays the character named **William Walker**. However, his real name is **Kanye Omari West**. The movie is a biographical drama based on his life, focusing on his struggles and perseverance.

