# OpenReview forum: "Benchmarking and Rethinking Knowledge Editing for Large Language Models"
_ICLR.cc/2026/Conference — Submitted to ICLR 2026_

### Official Review · Reviewer_fsgz · 2025-10-29

**Soundness:** 3
**Presentation:** 3
**Contribution:** 2
**Rating:** 4
**Confidence:** 4

**Summary:**

This paper presents a comprehensive benchmarking study of knowledge editing methods for large language models (LLMs), evaluating both parameter-modifying and non-parameter-modifying approaches under realistic settings. The authors introduce a simple yet strong baseline—Selective Contextual Reasoning (SCR)—and compare it against 12 recent methods across multiple datasets, model types, and editing scenarios. The results show that parameter-based editing methods often fail under sequential or complex editing settings, while context-based methods like SCR perform more robustly.

**Strengths:**

(1) The benchmarking is extensive and practical, covering autoregressive inference, sequential editing, event-level knowledge, and reasoning-oriented LLMs.
(2) The introduction of SCR as a strong and simple baseline is valuable and highlights the effectiveness of retrieval-augmented in-context learning.
(3) The evaluation includes multiple dimensions (reliability, generalization, locality, portability) and considers both general and reasoning-specific LLMs.
(4) The study raises important questions about the practicality of parameter-based editing methods in real-world scenarios.

**Weaknesses:**

(1) The contribution is somewhat incremental, as the benchmarking is largely conducted on existing datasets and follows established evaluation protocols without introducing new datasets or evaluation paradigms.
(2) While the extension to reasoning-oriented LLMs is a plus, the evaluation directly reuses existing tests rather than developing specialized metrics or settings tailored to reasoning capabilities.
(3) The study primarily experiments with smaller-scale models (e.g., 7B–14B parameters), which may limit the generalizability of findings to larger, state-of-the-art LLMs.

**Questions:**

What is the difference between SCR and In-Context Learning-based approaches？

---

> ### Author Response · Authors · 2025-12-03
> **Response to Reviewer fsgz**
>
> We sincerely thank you for recognizing the strengths of our work, which is very encouraging for us. We would like to take this opportunity to address your concerns.
>
> > W1: Incremental novelty
>
> We would like to emphasize that the primary contribution of our work lies in delivering a **comprehensive, practice-oriented evaluation** of knowledge editing methods. In addition to integrating multiple evaluation dimensions, **we introduce event-level datasets and extend the assessment to reasoning-oriented LLMs, areas that have not been systematically studied in prior work.** We believe this holistic and practical approach offers valuable insights for both researchers and practitioners and represents the core contribution of our paper.
>
> > W2: Evaluation on reasoning LLMs and metrics
>
> To maintain consistency across comparisons, we did not include reasoning-specific metrics such as reasoning length or semantic consistency. Nevertheless, we aim to emphasize the importance of a systematic and comprehensive evaluation for readers. We also discuss, in our results, the issues that arise in reasoning when errors occur.
>
> > W3: Model scale
>
> **We acknowledge that our experiments focus on models in the 7B–14B parameter range due to computational constraints.** While this is a limitation, we believe that the observed trends are likely to generalize to larger models, and our framework can be readily applied to larger-scale experiments when more resources are available in the future.
>
> > Q1: Difference between SCR and standard In-Context Learning
>
> In this work, SCR can be viewed as retrieval-augmented in-context learning (ICL). **Standard ICL assumes that all necessary knowledge is already provided in the context (an idealized scenario).**

---

### Official Review · Reviewer_BWjW · 2025-10-29

**Soundness:** 3
**Presentation:** 3
**Contribution:** 2
**Rating:** 4
**Confidence:** 3

**Summary:**

This paper presents a comprehensive benchmark of knowledge editing methods for LLMs across five families. The authors argue that prior evaluations over-emphasize on fact-level edits and overlook portability/generalization. They propose a unified, more realistic setup, e.g., including event-level and general-purpose datasets beyond fact triples, evaluating instruct and reasoning-oriented LLMs, and using autoregressive inference instead of teacher-forcing. They also introduce a simple baseline, i.e., selective contextual reasoning (SCR), which retrieves relevant knowledge text and injects it into the prompt. Empirically, parameter-modification methods underperform or collapse under sequential edits, while context-based approaches (IKE/ICE) and SCR are substantially more stable.

**Strengths:**

+ I agree that autoregressive decoding, sequential edits, and portability are all important and often ignored, and it's good to see that the paper productively consolidates them in one framework.
+ The author conducted extensive experiments that evaluates 12 recent methods across multiple LLMs (instruct + reasoning) and dataset types (facts, event-level; plus general reasoning benchmarks).
+ It's also interesting to see that that editing can harm reasoning capabilities.

**Weaknesses:**

- Although the paper provides some empirical insights, the core advance is an evaluation protocol + dataset selection and a simple SCR baseline. There is no new algorithmic insight for knowledge editing itself, which leads to limited novelty.

- The scale of edits seem to be limited. The experiments are conducted on 1 or 100 edits, but it seems that mass editing method such as Memit claims that they can scale up to thousands of edits. This raises concerns on whether the conclusions hold for larger-scale editing tasks.

- The conclusion suggests relying on in-context learning for small updates or re-training for large volumes, but it would be great if the author can have more in-depth discussions on how knowledge editing methods can evolve for more effective small-scale update.

**Questions:**

Please refer to my summary of weaknesses.

---

> ### Author Response · Authors · 2025-12-03
> **Response to Reviewer BWjW**
>
> We sincerely appreciate your positive feedback on our work, which is very encouraging. We would like to use this opportunity to respond to your concerns.
>
> > W1: Novelty / lack of model innovation
>
> As our submission was to the **Datasets and Benchmarks** track, the main contribution is to provide a systematic, comprehensive, and practice-oriented evaluation of existing knowledge editing methods, rather than proposing a new model. The SCR baseline we introduce is intentionally simple and intuitive, serving mainly as a reference point for comparison. Our goal is to provide a practical benchmark that can guide future research, and we hope that subsequent studies can include similar comparison baselines.
>
> > W2: Scale of edits
>
> In our experiments, we consider up to the full size of each dataset as the number of edits, as shown on the horizontal axis in Figure 2. Regarding MEMIT, its performance on Llama-3.1-8B-Instruct was below expectations, so we did not report it. However, on other LLMs (as shown in **Appendix Figures 4 and 5**), we observe that after more than 10 sequential edits, MEMIT’s performance drops to near zero across all evaluation dimensions.
>
> > W3: Discussion on small-scale updates
>
> In practice, updating model parameters, whether for large-scale or small-scale edits, remains costly and potentially risky. Looking forward, we envision approaches similar to PEFT methods that minimize the impact on global parameters, while incorporating additional constraints or regularization mechanisms during updates to improve stability and reliability.

---

### Official Review · Reviewer_hQHD · 2025-10-31

**Soundness:** 3
**Presentation:** 3
**Contribution:** 1
**Rating:** 2
**Confidence:** 3

**Summary:**

This paper presents a comprehensive benchmarking study of knowledge editing methods for LLMs. The authors evaluate recent methods across on various LLMs, datasets, and under practical settings like autoregressive inference and sequential editing. They also compare those methods against Selective Contextual Reasoning (SCR). The paper's key findings suggest that parameter-modification-based editing methods perform poorly in realistic scenarios, often degrading the model's reasoning abilities and failing under sequential edits. In contrast, the SCR baseline consistently outperforms these methods, demonstrating robustness and effectiveness. The authors conclude that for knowledge updates, selectively injecting external knowledge into the context is a more robust strategy than modifying model parameters, especially when dealing with a limited number of updates.

**Strengths:**

1. This paper is clearly written, well organized, and generally easy to understand.

2. This paper addresses the important problem of knowledge editing in LLMs.

3. The authors' effort in conducting a large-scale, comprehensive benchmark is commendable.

**Weaknesses:**

The primary weakness of this paper is its limited novelty. While the benchmarking effort is extensive, the core research questions and many of the conclusions align closely with existing knowledge and prior work, making the contribution more of a confirmation than a new discovery.

1. RQ1 investigates editing under autoregressive inference and sequential editing scenarios. The challenges of sequential editing are well-investigated problems in the field. Similarly, previous works such as [1], [2] and [3] have already adopted autoregressive decoding in their evaluations, moving away from the less practical teacher-forcing paradigm. Therefore, the findings in this section, which show that parameter-modification methods struggle under these conditions, are largely confirmatory.

2. RQ3 explores editing more complex, event-level knowledge. However, the conclusion that parameter-modification methods struggle to handle the complex semantic relationships inherent in events is an expected extension of findings from fact-based editing. The results are similar to those observed in [1] and [2].

3. RQ4 investigates the trade-off between edit time and inference latency. It is intuitive that methods involving parameter updates would have higher edit-time costs but no additional inference overhead. Conversely, context-based methods that augment the input prompt naturally increase inference latency due to longer sequence processing. This conclusion does not offer new insights.

[1] Everything is Editable: Extend Knowledge Editing to Unstructured Data in Large Language Models

[2] AnyEdit: Edit Any Knowledge Encoded in Language Models

[3] MQuAKE: Assessing Knowledge Editing in Language Models via Multi-Hop

**Questions:**

See weaknesses.

---

> ### Author Response · Authors · 2025-12-03
> **Response to Reviewer hQHD**
>
> We sincerely thank you for your careful and constructive feedback. We are glad that you recognize the clarity and comprehensiveness of our work. We respond to the main concerns below.
>
> > W1 & W2 & W3: The contribution is more of a confirmation than a new discovery
>
> While prior works have explored individual aspects, such as autoregressive inference and sequential editing, these studies are often fragmented, limited in scope, and evaluated under narrow settings. **Conclusions from such studies may hold in isolated scenarios, but cannot reliably inform practitioners or readers about realistic model behavior.** The continued publication of methods without considering these practical settings illustrates the need for a holistic and systematic evaluation, which our work provides.
>
> We would also like to highlight that **RQ2** remains an important contribution of our paper. Specifically, we analyze the impact of knowledge editing on reasoning LLMs, showing that parameter-modification-based methods can significantly degrade reasoning ability. Additionally, we examine editing of more complex knowledge (event-level information) (**RQ3**) and efficiency trade-offs (**RQ4**), which are critical practical considerations that have been largely overlooked in prior studies. These analyses provide actionable insights for both researchers and practitioners, reinforcing the relevance and importance of our benchmark beyond merely confirming known effects.
>
> We would like to emphasize that the primary goal of our work is not to introduce a new editing model, but to provide a systematic, comprehensive, and practice-oriented evaluation of existing knowledge editing methods for large language models (LLMs).

---

### Official Review · Reviewer_Y1yQ · 2025-11-01

**Soundness:** 2
**Presentation:** 3
**Contribution:** 2
**Rating:** 4
**Confidence:** 3

**Summary:**

This paper conducts a comprehensive and unified benchmark of knowledge editing methods for large language models (LLMs), addressing the inconsistency and unrealistic assumptions in prior work. It evaluates twelve representative approaches across five paradigms—parameter-based, meta-learning, adapter-based, in-context, and external-memory—using diverse datasets spanning factual, event-level, and reasoning tasks. The authors introduce four evaluation dimensions (reliability, generalization, locality, and portability) and propose a simple yet strong baseline, Selective Contextual Reasoning (SCR), which retrieves and injects relevant knowledge into prompts. Experiments on multiple LLMs (Llama2/3, Mistral, DeepSeek-R1) reveal that parameter-modification methods collapse under sequential edits and harm reasoning ability, while SCR and other context-based methods remain stable and robust. The study concludes that context-driven knowledge integration is a more practical and reliable approach than parameter editing for real-world LLM updates.

**Strengths:**

1. This paper is good writing and easy to follow. The idea and findings are presented very clear.
2. This paper conduct comprehensive experiments covering different types of base models, editing methods and benchmarks.
3. The findings are promising and reasonable.

**Weaknesses:**

1. The technical and even the practical contribution is limited. This paper didn't provide theoretical analysis why knowledge editing models hurt the model's capability or explain why sequential editing will fail. Second, this paper didn't give any technical contribution in how to mitigating the side effects of knowledge editing. Third, although the paper claims that they "aim to conduct a comprehensive benchmark study", they only use existing datasets for evaluation, instead of making any contribution for building a benchmark. Although the paper introduce the SRC, the main idea of SRC is too simple which is not technical enough for a published paper, or the improvement is not significant based on Table 1/2/3 results. To summary, this paper only conduct evaluation experiments and summarizes some findings, lacking of deep analysis and significant contribution.
2. In line 95, this paper claims that "Regarding the editing scenarios, many studies do not test in sequential / continuous editing scenario", however, there are many knowledge editing work that conduct sequential editing experiments: (1) Navigating the Dual Facets: A Comprehensive Evaluation of Sequential Memory Editing in Large Language Models (https://arxiv.org/abs/2402.11122); (2) Model Editing at Scale leads to Gradual and Catastrophic Forgetting (https://arxiv.org/pdf/2401.07453). Both paper not only conduct comprehensive analysis, but also provides analysis and explanation of why the side effects happens and how to solve it (or provide some potential direction). It would be better to incorporate those papers into related work, and clarify the differences between your work and those related work.

**Questions:**

I have a question about SCR: how sensitive is SCR’s performance to retrieval quality (e.g., noisy or incomplete knowledge bases)? How do you make sure the retrieval is correct to make sure the promising of this method?

---

> ### Author Response · Authors · 2025-12-03
> **Response to Reviewer Y1yQ**
>
> We sincerely appreciate your detailed and constructive feedback. We are pleased that you found our work clear, comprehensive, and practically meaningful. Below we respond to all concerns point by point.
>
> > W1: Limited technical and practical contributions
>
> This paper is submited to the **Datasets and Benchmarks** track. Our work is not intended to make a model-design technical contribution, and SCR is introduced as a reference baseline rather than being emphasized as an innovative model. Instead, we believe that a comprehensive, practice-oriented evaluation is what the field urgently needs at this stage. This is crucial for guiding both future research and real-world deployment.
>
> We evaluate these methods across various perspectives: the editing content (facts and events), LLM type (general LLMs and reasoning LLMs), inference setting (autoregressive generation instead of the commonly used teacher-forcing decoding) and evaluation dimensions of editing effectiveness (including four key dimensions with a particular emphasis on portability, which is often overlooked). Additionally, we assess the model's performance in terms of both editing outcomes and unrelated task capabilities.
>
> > W2: Missing related references
>
> In the original paper, "many" was used rather than "all". Indeed, while some previous works have addressed sequential editing, **there are still studies that do not consider it.** **More importantly, none of the existing works evaluate multi-editing from such a comprehensive and practical perspective, which we believe is our core contribution.** We will incorporate these works and expand the related work discussion accordingly.
>
> > Q1: Sensitivity of SCR to retrieval quality
>
> SCR is a simple framework rather than a novel editing method. **In practical deployment, SCR can rely on the latest or domain-optimized retrievers, but no retrieval system can guarantee 100% correctness.** We introduce SCR simply for comparison with parameter-modification-based editing methods, and to encourage future studies to consider such parameter-free comparison baselines.

---

### Author Response · Authors · 2025-12-03
**Response to AC**

Dear AC,

We sincerely thank the reviewers for their constructive feedback and for recognizing the clarity, comprehensiveness, and practical insights of our work. Our evaluation considers multiple perspectives, including the **editing content** (facts and events), **LLM type** (general and reasoning-oriented models), **inference setting** (autoregressive generation instead of the commonly used teacher-forcing decoding), **number of edits** (single and sequential edits), and **several dimensions of editing effectiveness**, with particular emphasis on **portability**, which is often overlooked. We further assess the models' performance not only on the edited knowledge but also on **unrelated tasks** to measure overall robustness. In addition, we introduce a **simple and intuitive baseline, SCR, for comparison**, and discuss the **efficiency and applicability** of different types of editing methods. Below, we clarify the reviewers' concerns regarding **novelty** and address some additional **technical points**. We believe that our **comprehensive, practice-oriented benchmarking** is well-suited for the **Datasets and Benchmarks track** and provides valuable guidance for the community.

Authors

---

### Meta-Review · Area_Chair_rwW8 · 2026-01-02

**Summary:**

**Paper Summary:**

The paper presents a comprehensive benchmark of knowledge editing methods for large language models, evaluating parameter-based and context-based approaches under realistic settings and introducing a simple retrieval-augmented baseline.

**Strengths:**
1. The paper addresses an important problem in LLM knowledge editing.
2. The benchmark provides a large-scale, systematic evaluation across multiple dimensions (reliability, generalization, locality, portability).
3. The benchmark includes diverse datasets (fact-level, event-level, reasoning tasks) and practical scenarios (autoregressive inference, sequential edits).
4. It introduces a simple yet strong baseline (Selective Contextual Reasoning) that demonstrates robustness compared to parameter-modification methods.

**Weaknesses:**
1. Limited novelty: mainly an evaluation study without new algorithmic contributions/datasets. There are also similar previous works that have not been cited and compared in the paper.
2. The paper relies on existing datasets and does not introduce new benchmarks or theoretical insights.
3. The findings are largely confirmatory of prior work; it lacks deep analysis of why failures occur or how to mitigate side effects.
4. The experiments focus on small-scale edits and mid-sized models (7B–14B), limiting generalizability to large-scale scenarios.
5. The SCR baseline is simple and lacks technical depth; the improvement over existing methods is modest.

**Reviewer Concerns:**

**Reviewer Y1yQ**

Addressed:
1. Clarified that the paper is submitted to the Datasets and Benchmarks track, so lack of algorithmic novelty is expected.
2. Explained that SCR is intended as a simple baseline, not a technical contribution.
3. Acknowledged missing related work and promised to incorporate additional references.

Outstanding:
1. No theoretical explanation for why sequential editing fails or why reasoning ability degrades.
2. No proposed mitigation strategies for side effects of knowledge editing.
3. Benchmark still relies on existing datasets; no new dataset contribution.

**Reviewer hQHD**

Addressed:
1. Justified the need for a holistic evaluation, arguing prior studies are fragmented and narrow.
2. Highlighted contributions such as analyzing reasoning LLMs and event-level edits.

Outstanding:
1. Findings remain largely confirmatory; rebuttal does not introduce new insights beyond framing the scope.
2. No new evaluation paradigms or metrics for reasoning tasks.
3. Novelty concerns persist.

**Reviewer BWjW**

Addressed:
1. Clarified that novelty lies in systematic benchmarking, not model innovation.
2. Discussed scale of edits and provided additional observations on MEMIT performance.
3. Offered forward-looking ideas for small-scale updates (e.g., PEFT-like approaches).

Outstanding:
1. Limited discussion on large-scale editing beyond acknowledging constraints.
2. No concrete roadmap for evolving knowledge editing methods.

**Reviewer fsgz**

Addressed:
1. Emphasized event-level datasets and reasoning LLMs as contributions.
2. Explained consistency in evaluation metrics and acknowledged resource constraints for model scale.
3. Clarified SCR as retrieval-augmented in-context learning.

Outstanding:
1. Incremental nature of contribution remains; no new datasets or specialized reasoning metrics.
2. Generalizability to larger models is still uncertain.

**Reviewer Scores:**

While the authors have provided some clarifications, many critical questions have not been addressed in the rebuttal. Therefore it's difficult to judge whether the reviewers would like to change their scores.

---

### Decision · Program_Chairs · 2026-01-26

Reject